

# Differences in depression prevalence among older adults in China before and during the COVID-19 pandemic: a systematic review and meta-analysis

Xin Zhao[1], Xiaojing Du[2], Shuliang Bai[1], Pianpian Zheng[3], Xun Zhou[4] and Zhenjie Wang[5]

[1] Nanjing University, Nanjing, China
[2] Xi'an Jiaotong University, Xi'an, China
[3] Jiangxi University of Finance and Economics, Nanchang, China
[4] Beijing Institute of Graphic Communication, Beijing, China
[5] Peking University, Beijing, China

Corresponding author
Zhenjie Wang,
zhenjie.wang@pku.edu.cn

## ABSTRACT

**Objective**. Changes in the prevalence of depression during the COVID-19 (Coronavirus disease 2019) pandemic among older adults in China have not been systematically evaluated. We aimed to systematically summarize existing evidence to conduct a meta-analysis to quantify changes in the prevalence of depression before and during the COVID-19 pandemic.

**Methods**. The PubMed, Web of Science, Scopus, Embase, PsycINFO, China National Knowledge Infrastructure (CNKI), WangFang Data, CQVIP, and China Biology Medicine disc (CBMdisc) databases were searched from January 1, 2017, to November 20, 2024. Studies reporting the prevalence of depression among Chinese individuals aged 60 or older using validated diagnostic tools were included. A random-effects model was applied to estimate pooled prevalence, with subgroup analyses performed by demographic and socio-economic factors. Relative risks (RR) were calculated to compare prevalence across different pandemic stages.

**Results**. A total of 101 studies involving 264,758 participants were included. The pooled prevalence of depression among older adults was 25.8% (95% CI [21.7–29.9]) from 2017–2019 and 23.8% (95% CI [19.8–27.8]) from 2020–2023. During the early pandemic stage (January–April 2020), prevalence significantly decreased (RR = 0.849, $P < .001$) but increased in later stages, reaching 24.4% by 2021–2023. The prevalence of depression among older adults during the COVID-19 pandemic showed a significant increasing trend ($P_{trend} < .001$). Subgroup analyses revealed higher prevalence among females, rural residents, individuals with lower education levels, and those living alone.

**Conclusion**. Depression prevalence among older adults in China decreased during the early pandemic response but showed an increasing trend over time, reflecting the complex mental health impact of prolonged public health measures. Effective interventions are needed to address the specific needs of vulnerable subgroups during and beyond public health crises.

## INTRODUCTION

The World Health Organization (WHO) defines depressive disorder as a common mental health condition. It involves long periods of low mood or loss of interest in activities that were once enjoyable (*World Health Organization (WHO), 2023a*). It affects mental well-being and makes social interactions harder (*Steger & Kashdan, 2009*). It can also lead to cognitive decline (*Djernes, 2006*; *Vink, Aartsen & Schoevers, 2008*), sleep problems (*Murphy & Peterson, 2014*; *Fang et al., 2019*), and difficulties in daily activities (*Baune et al., 2010*; *Lin et al., 2014*). It is estimated that almost one in eight people globally live with a mental illness, and nearly one in two people will experience mental illness at some point in their lifetime (*World Health Organization (WHO), 2022*). Recent data from WHO (*World Health Organization (WHO), 2023a*) indicated that approximately 280 million people worldwide are affected by depression.

Depression in older adults results from a combination of social, psychological, and biological factors (*Sidik, Rampal & Afifi, 2004*; *Suwanmanee et al., 2012*). It increases the risk of chronic illnesses, leads to greater healthcare use, and reduces quality of life (*Chapman & Perry, 2008*; *Luppa et al., 2012*). Depression among older adults is a significant global public health concern (*El-Gilany, Elkhawaga & Sarraf, 2018*; *World Health Organization (WHO), 2023b*). The prevalence of depression in older adults varies widely, ranging from 10% to 55% (*Roh et al., 2015*). In China, the estimated prevalence of depression in older adults ranges from 5.6% (*Wang et al., 2022b*) to 75.4% (*Mao, Lu & Xiao, 2022*). A study based on the Global Burden of Disease indicated that the prevalence of depression among older adults in China showed an increasing trend from 1990 to 2017 (*Ren et al., 2020*). A recent meta-analysis revealed that the prevalence of depression was 20.0% in China (*Tang, Jiang & Tang, 2021*).

Coronavirus disease 2019 (COVID-19) has had a profound impact on the mental health and well-being of people worldwide (*United Nations (UN), 2020*). Additionally, the public health measures taken to contain the spread of the virus have had a significant effect on the mental health of individuals (*Arora et al., 2022*; *Mahmud et al., 2021*). Although many people have adapted to the pandemic (*Pierce et al., 2021*), many people still experience mental health problems. Mental health issues among older people are often underrecognized and undertreated (*Rodda, Walker & Carter, 2011*; *Herrman et al., 2022*), as the clinical manifestations of these issues are often attributed to normal aging, loss or physical illness (*Herrman et al., 2022*). The WHO reported that the global prevalence of anxiety and depression increased dramatically (by 25%) during the first year of the COVID-19 pandemic (*World Health Organization (WHO), 2022*). This increase significantly impacted older adults, as isolation, health concerns, and fear of illness and death contributed to a higher prevalence of depression. In China, the prevalence of depression among older adults was 26.4% during the pandemic (*Guo et al., 2020*). Another online survey in 2020 reported that the prevalence of depression was 30.8% among Chinese older adults (*Liang et al., 2021a*).

Many studies have examined depression in older adults. However, few have systematically examined how depression prevalence changed during different stages of the COVID-19

pandemic in China. Most studies provide cross-sectional estimates and do not track trends over time or consider the effects of different control measures. China used strict pandemic strategies, such as the dynamic zero-COVID policy and strict movement restrictions. A detailed review of existing research is needed to understand how depression prevalence changed before and during the pandemic. Unlike other countries that used milder strategies, China's strict but flexible approach may have affected older adults' mental health in different ways at different times.

This study fills the gap by analyzing changes in depression prevalence across the pandemic through a meta-analysis. This approach helps show how China's response affected older adults' mental health. The findings can guide future public health efforts to support mental well-being in this vulnerable group.

# MATERIALS AND METHODS

This study was conducted in accordance with the Preferred Reporting Items for Systematic Reviews and Meta-analyses (PRISMA) reporting guidelines (*Page et al., 2021*). This study was registered in PROSPERO under the registration number CRD42023402865.

## Search strategy

We conducted a systematic and rolling search across five English-language databases (PubMed, Web of Science, Scopus, Embase, PsycINFO) and four Chinese-language databases: China National Knowledge Infrastructure (CNKI), WangFang Data, CQVIP, and China Biology Medicine disc (CBMdisc). This rolling search was updated continuously to ensure inclusion of the most recent studies, with the final search conducted on November 20, 2024, covering studies published from January 1, 2017, to the final search date. The search terms used were keywords related to the older population (elder OR elderly OR "old population" OR aged OR "old adults" or "older adults" OR geriatrics OR "late life"), depression (depression OR depressive OR "depressive disorder" OR "depressive symptom" OR depressed OR "depressed mental" OR distress OR "mental distress" OR dysthymia OR "dysthymic disorder" OR "late-life depression"), and China (China OR Chinese).

The rationale for selecting 2017 as the starting point was to ensure a sufficient pre-pandemic baseline for analyzing trends in depression prevalence before the onset of COVID-19. The study period extends to 2024, as our systematic search aimed to incorporate the most recently published studies that met our inclusion criteria, allowing for a comprehensive analysis of depression prevalence trends before and during the pandemic. The actual data collection periods of the included studies are reported in the results section.

## Inclusion and exclusion criteria

The inclusion criteria were as follow: (a) articles published in Chinese or English between January 1, 2017, and November 20, 2024; (b) studies reporting the prevalence of depression with accessible data; (c) participants aged 60 years or above from China; (d) cross-sectional studies; (e) if multiple studies were published based on the same database, we only included

the study that provided the most comprehensive and relevant data for our analysis (*e.g.*, the largest sample size or the most detailed subgroup analysis); and (f) studies employing standardized and validated scales for depression measurement.

Exclusion criteria included: (a) reviews, conference papers, case reports, comments, letters, editorials, and unpublished manuscripts; (b) studies employing unscientific research designs, such as convenience sampling; (c) studies conducted on specialized populations, including individuals with chronic conditions, inpatients, those with specific diseases (*e.g.*, cancer, stroke, or HIV/AIDS), or individuals impacted by specific disasters or crises (*e.g.*, earthquakes); and (d) studies where full texts were unavailable through online databases, library services, or correspondence with authors.

## Study selection

Two authors (ZX and WZJ) independently searched all nine databases *via* the same search term to ensure that no relevant studies were overlooked. The screening procedure is outlined below: (a) the titles were reviewed to identify potential articles related to the topic, (b) the abstracts were reviewed to narrow the list of articles, and (c) the full texts of the articles were read to make a final decision. Two authors (ZX and WZJ) conducted a preliminary screening of the titles and abstracts of the database records. They then retrieved the full texts of the records for further assessment. Finally, they independently qualified the full-text records. The literature retrieved from nine databases was imported into Excel, and duplicates were removed.

The screening process was divided into two steps. The first step involved filtering by examining the title and abstract, and the second step involved filtering by browsing the full text. Two authors (ZX and WZJ) independently screened the titles and abstracts of the records in the databases, read the full texts and performed quality assessment. In the event of any discrepancies, a third party (DXJ and ZPP) was consulted to facilitate discussion and resolution.

## Quality assessment

The quality of the included studies was assessed by the 11-item checklist recommended by the Agency for Healthcare Research and Quality (AHRQ) (*Rostom et al., 2004*). The item was given a score of 1 if the answer was "yes" and a score of 0 if the answer was "no" or "unclear" (opposite for the 5th item). A total score of 0–3 = low quality, 4–7 = moderate quality, and 8–11 = high quality was given.

## Data extraction

After eligible studies were identified, a standardized form was used to extract the following data from each study: authors, publication year, survey period, survey location, diagnostic tools, sample size, age, and number of depression cases. Additionally, the prevalence of depression among older adults was collected and stratified by sex, education level, marital status, and living arrangements.

## COVID-19 stages in China

The Chinese government divided its COVID-19 response into five stages (*The State Council Information Office of the People's Republic of China (SCIO), 2020*): initial response

(December 27, 2019, to January 19, 2020), preliminary containment (January 20 to February 20, 2020), gradual reduction in new cases (February 21 to March 17, 2020), decisive victory in Wuhan, Hubei (March 18 to April 28, 2020), and regular epidemic prevention and control (after April 29, 2020).

In the present study, the development stages of COVID-19 were defined as follows: stage 1 (January–April 2020) was marked by the timing of Wuhan's lockdown and reopening; stage 2 (May–December 2020) represented the regular epidemic prevention and control phase; and stage 3 (2021–2023) encompassed the implementation of the dynamic zero-COVID-19 policy beginning in August 2021, with a significant easing of COVID-19 restrictions at the end of 2022.

## Data analysis

A random effects model was used to estimate the prevalence of depression among older adults in China, with 95% confidence intervals (CIs) representing the effect sizes of all studies. Sensitivity analysis was performed by excluding each study one at a time and recalculating the prevalence rates for the remaining studies to determine the robustness of the main results. Publication bias was assessed *via* funnel plots, Begg's test, and Egger's test, with all raw data logit-transformed.

Using the prevalence of depression from 2017–2019 or January to April 2020 as reference values, the relative risk (RR) for different pandemic stages was calculated sequentially (Supplementary Material). The chi-square test was used to examine differences in prevalence rates across these stages. The Mann–Kendall test was applied to evaluate trends in the prevalence of depression during the COVID–19 pandemic. Statistical analyses were performed *via* R version 4.3.2 and Stata 16.0. (Fig. S1).

## RESULTS

### Published evidence characteristics

A total of 101 studies involving 264,758 participants (63,063 participants with depression) were included in this meta-analysis (Fig. 1). Considering that all studies included in this research were published between 2017 and November 20, 2024, but all data collection occurred between 2017 and 2023, we divided the studies into two groups based on the period of data collection: the pre-pandemic period (2017–2019) and the pandemic period (2020–2023). All subsequent analyses in this study were conducted based on this grouping. The quality of the included studies was assessed *via* the AHRQ checklist, indicating that all included studies were of moderate or high quality (Tables S1 and S2).

### Differences before and during the COVID-19 pandemic

From 2017 to 2019, 51 studies (*Wang et al., 2022b*; *Li et al., 2022e*; *Li et al., 2021a*; *Yang et al., 2021b*; *Zhang et al., 2021a*; *Chen et al., 2019*; *Wang et al., 2022a*; *Liu et al., 2022c*; *Wang et al., 2021a*; *Han & Shi, 2021*; *Sun et al., 2020*; *Ma et al., 2020*; *Ding et al., 2018*; *Liu et al., 2021d*; *Li et al., 2021b*; *Zhang et al., 2021b*; *Zhai et al., 2023*; *Yan et al., 2022*; *Peng et al., 2022*; *Hou et al., 2022a*; *Zhao et al., 2022*; *Gao, Hu & He, 2022*; *Xu et al., 2022*; *Jiang et al., 2022a*; *Jiang et al., 2022b*; *Ding et al., 2022b*; *Zhang et al., 2022*; *Hu et al., 2022*; *Rong et al.,*

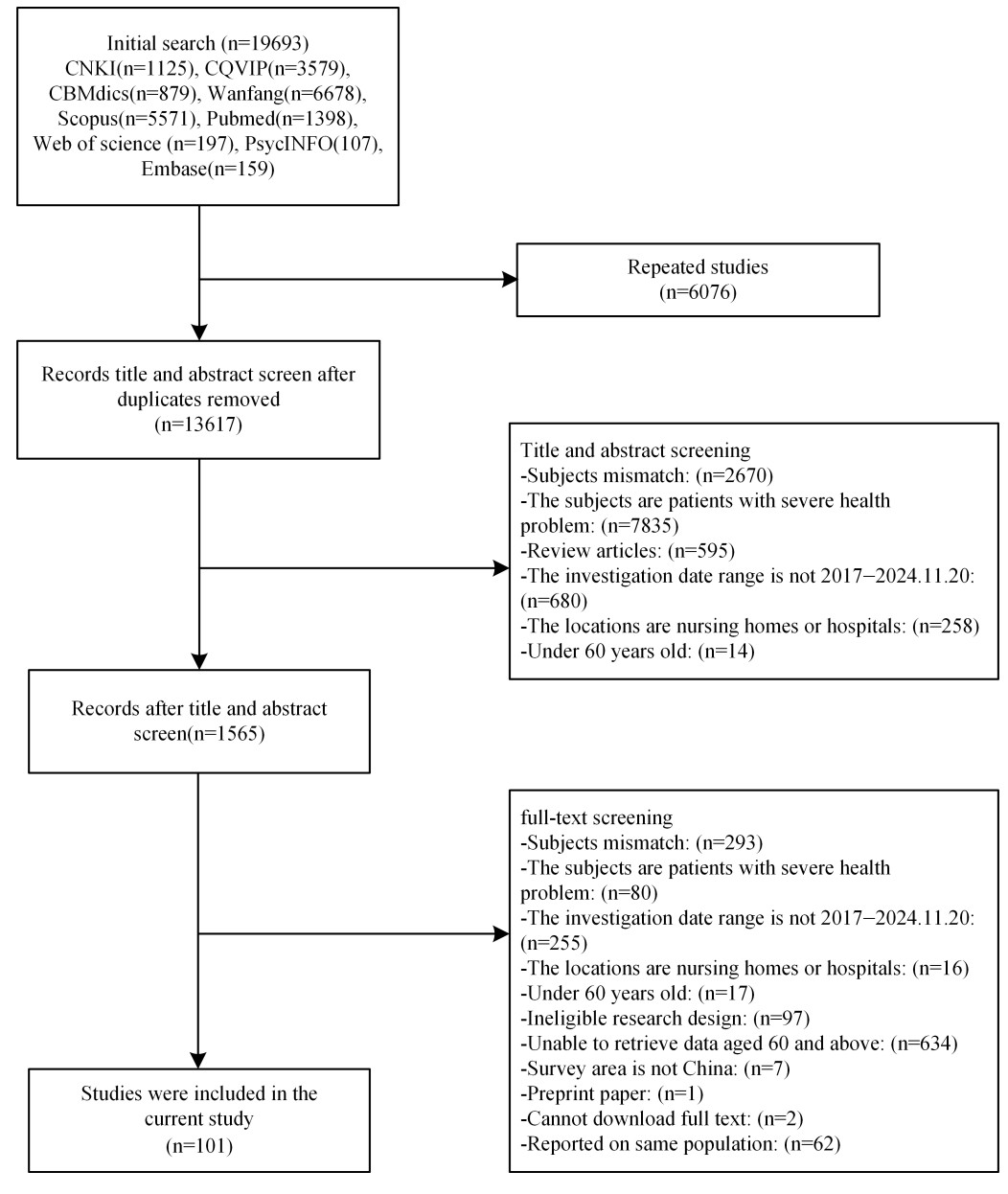

**Figure 1  PRISMA flowchart.**

*2021*; *Lin et al., 2021*; *Liang et al., 2021c*; *Liu et al., 2021c*; *Yang et al., 2021a*; *Qiu et al., 2020*; *Gu et al., 2020*; *Chu et al., 2020*; *Wu et al., 2020a*; *Lin et al., 2020*; *Dai et al., 2019*; *You et al., 2022*; *Zhao et al., 2020*; *Li et al., 2023*; *Li et al., 2022a*; *Yuan et al., 2024*; *Xiong et al., 2022*; *Guo et al., 2022*; *Xiong et al., 2023*; *Chang et al., 2023*; *Zeng et al., 2023*; *Wang et al., 2024*; *Wang et al., 2023b*) involving 171,667 participants reported 41,473 cases of depression, resulting in a pooled prevalence of 25.8% (95% CI [21.7–29.9]). From 2020 to 2023, 50 studies (*Mao, Lu & Xiao, 2022*; *Liang et al., 2021a*; *Chen et al., 2023*; *Qin, Deng & Li, 2022*; *Yu et al., 2022*; *Qin et al., 2022*; *Liu et al., 2022b*; *Cui, Dong & Yang, 2022*; *Xu, 2022*; *Liu*

*et al., 2022a; Lu et al., 2022; Lu et al., 2023; Hao et al., 2023; Li, Lin & Wu, 2022; Li et al., 2022c; Li et al., 2022d; Ding et al., 2022a; Hou et al., 2022b; Liu et al., 2022d; Wang et al., 2021b; Liu et al., 2021a; Zhou et al., 2021; Wang & Tang, 2020; Wu et al., 2020b; Wang et al., 2020b; He et al., 2022; Li et al., 2022b; Yu et al., 2024; Zhao et al., 2024; Lin et al., 2024; Zong & Ge, 2023; Zhang et al., 2024a; Liu et al., 2023; Yang et al., 2024; Liu, Li & Sun, 2024; Huang & Qiu, 2024; Zhao et al., 2023; Li et al., 2024; Su et al., 2023; Lin et al., 2023; Yang et al., 2023; Ge et al., 2023; Jiang et al., 2024; Zhang et al., 2024b; Hu et al., 2023; Gan, Yao & Li, 2024; Xiong et al., 2024; Guo et al., 2024; Peng et al., 2024; He et al., 2024*) involving 93,091 participants reported 21,590 cases of depression, with a pooled prevalence of 23.8% (95% CI [19.8–27.8]). The difference between these pooled prevalence estimates was statistically significant ($P < .001$).

To further examine depression prevalence during the active phase of COVID-19, we analyzed studies with data collected from 2020 to 2022, which showed a pooled prevalence of 23.1% (95% CI [19.2–27.0]). Additionally, among studies with data collected in 2023 ($n = 3$), the estimated prevalence was 33.9% (95% CI [20.3–47.6]).

Figure 2 displays the results of the comparative analysis of the prevalence of depression among older adults for two periods: 2017 to 2019 and 2020 to 2023 (Table S3). Compared with 2017–2019, there was a decrease in the prevalence of depression among both males and females in China from 2020–2023. This increasing trend was observed across various age subgroups, including those aged 60 to 69, 70 to 79, and over 80 years. The prevalence of depression decreased in rural areas but increased in urban areas. In terms of educational level, there was an increase in the prevalence of depression among older adults with primary school or junior high school education, whereas a slight decrease was observed among those with senior high school education or higher. Additionally, the prevalence of depression increased among older adults living alone, whereas it decreased among those not living alone.

## Effects of COVID-19 on the prevalence of depression

In the period from January to April 2020, the prevalence of depression among older adults in China was 21.9%, representing a 15.1% decrease from the 25.8% reported for 2017–2019 (RR = 0.849, $P < .001$). From May to December 2020, the prevalence increased to 23.9%, which was 7.4% lower than the 2017–2019 level, although this difference was not statistically significant (RR = 0.926, $P = .338$). In the period from 2021 to 2023, the prevalence rose to 24.4%, 5.4% lower than the 2017–2019 level (RR = 0.946, $P < .001$) (Fig. 3). The risk of depression increased by 9.1% from January–April 2020 to May–December 2020 (RR = 1.091, $P < .001$) and by 11.4% from January–April 2020 to 2021–2023 (RR = 1.114, $P < .001$) (Fig. 4). A significant increasing trend in depression prevalence among older adults was observed during the pandemic period ($P_{trend} < .001$) (Fig. 5).

## Subgroup analysis

The results of the subgroup analysis (Table 1) revealed significant variability in depression prevalence across sex, age group, living area, education level, marital status, and residential status among Chinese older adults. In the pre-pandemic period (2017–2019), depression

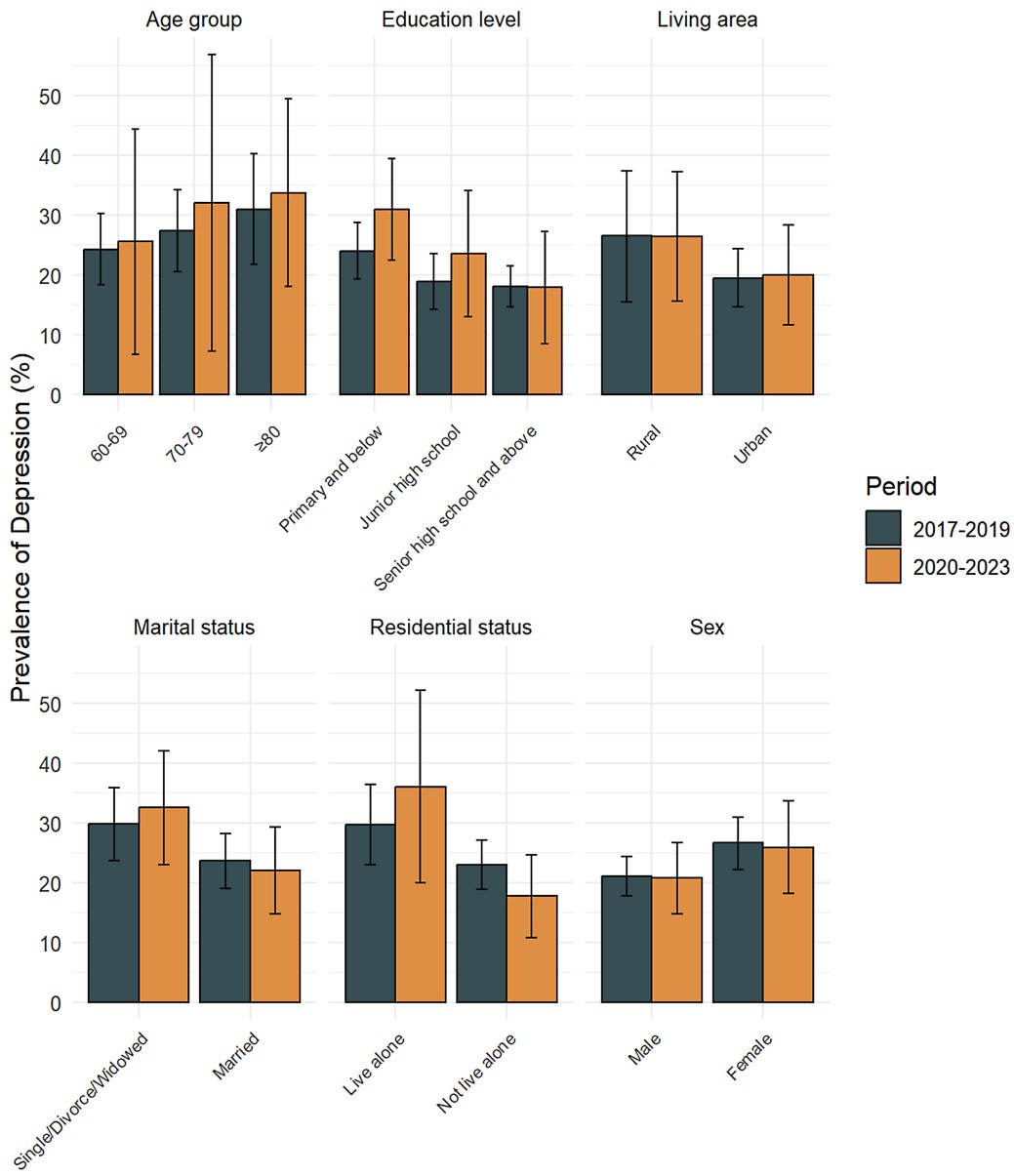

**Figure 2   Demographic differences in the prevalence of depression among older adults before and during the COVID-19 pandemic 2017–2019, 2020–2023.**

prevalence differed significantly across all subgroup variables, including sex, age group, living area, education level, marital status, and residential status (all $P < .001$). In the pandemic period (2020–2023), significant differences persisted for sex, living area, education level, marital status, and residential status (all $P < .001$).

Throughout both periods, depression prevalence remained consistently higher among older women than older men, rural than urban residents, and individuals with lower educational levels (primary school or below). Additionally, depression prevalence was
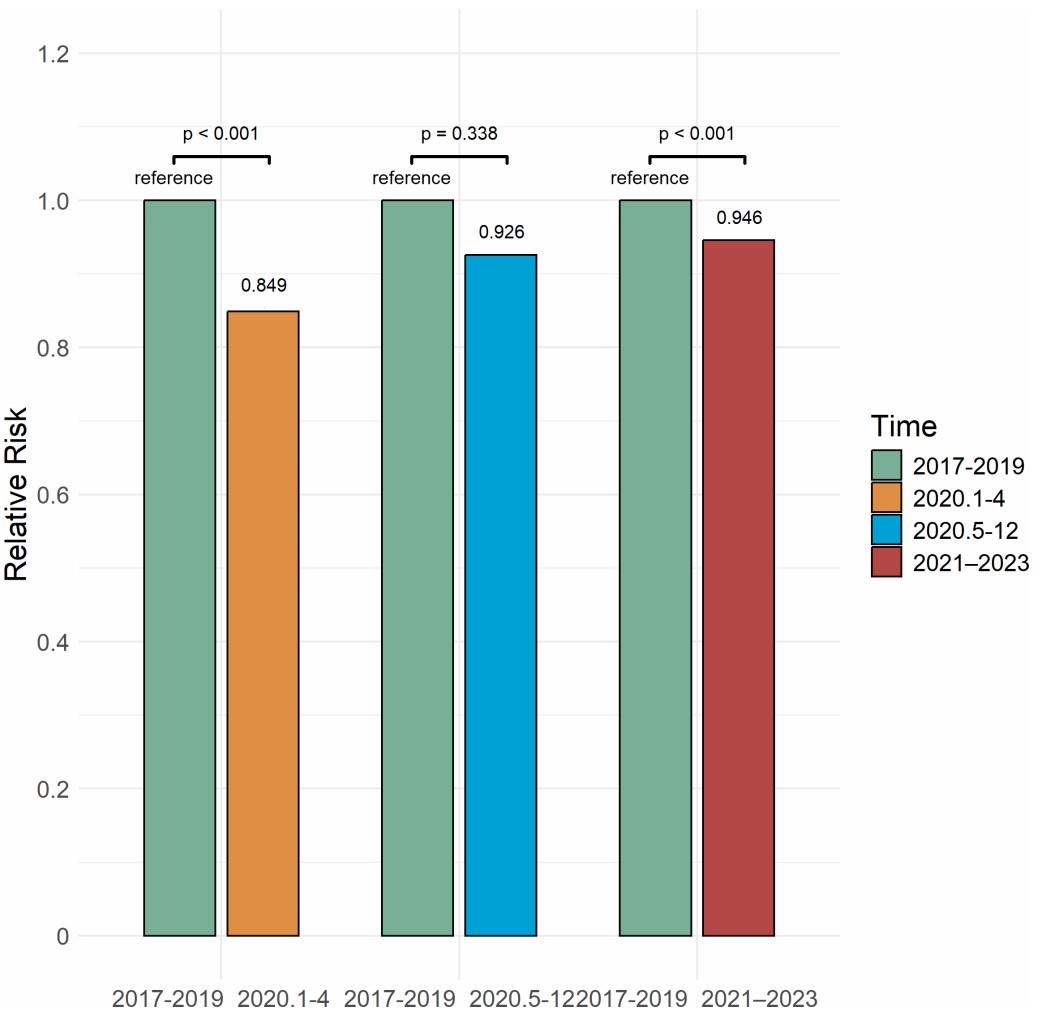

**Figure 3** Relative risk of depression among Chinese older adults 2017–2023.

higher among single, divorced, or widowed individuals compared to their married counterparts, and among those living alone compared to those living with others.

## Sensitivity analysis

Begg's test showed that there was no obvious publication bias (2017–2019: $P = .27$; 2020–2023: $P = .74$), and Egger's test also indicated that there was no obvious publication bias (2017–2019: $P = .73$; 2020–2023: $P = .47$).

The sensitivity analysis results for studies from 2017 to 2019 ranged between 24.9% (95% CI [21.2–28.6]) and 26.8% (95% CI [21.8–30.3]) (Table S4), while for 2020 to 2023, the results ranged between 22.7% (95% CI [18.8–26.6]) and 24.2% (95% CI [19.9–28.6]) (Table S5), indicating that the primary results were robust. To further assess the robustness of our findings during the active phase of COVID-19 (2020–2022), we conducted an additional sensitivity analysis for this period. The results remained consistent, with a pooled prevalence ranging from 22.0% (95% CI [18.1–25.8]) to 23.6% (95% CI [19.2–27.9]) using

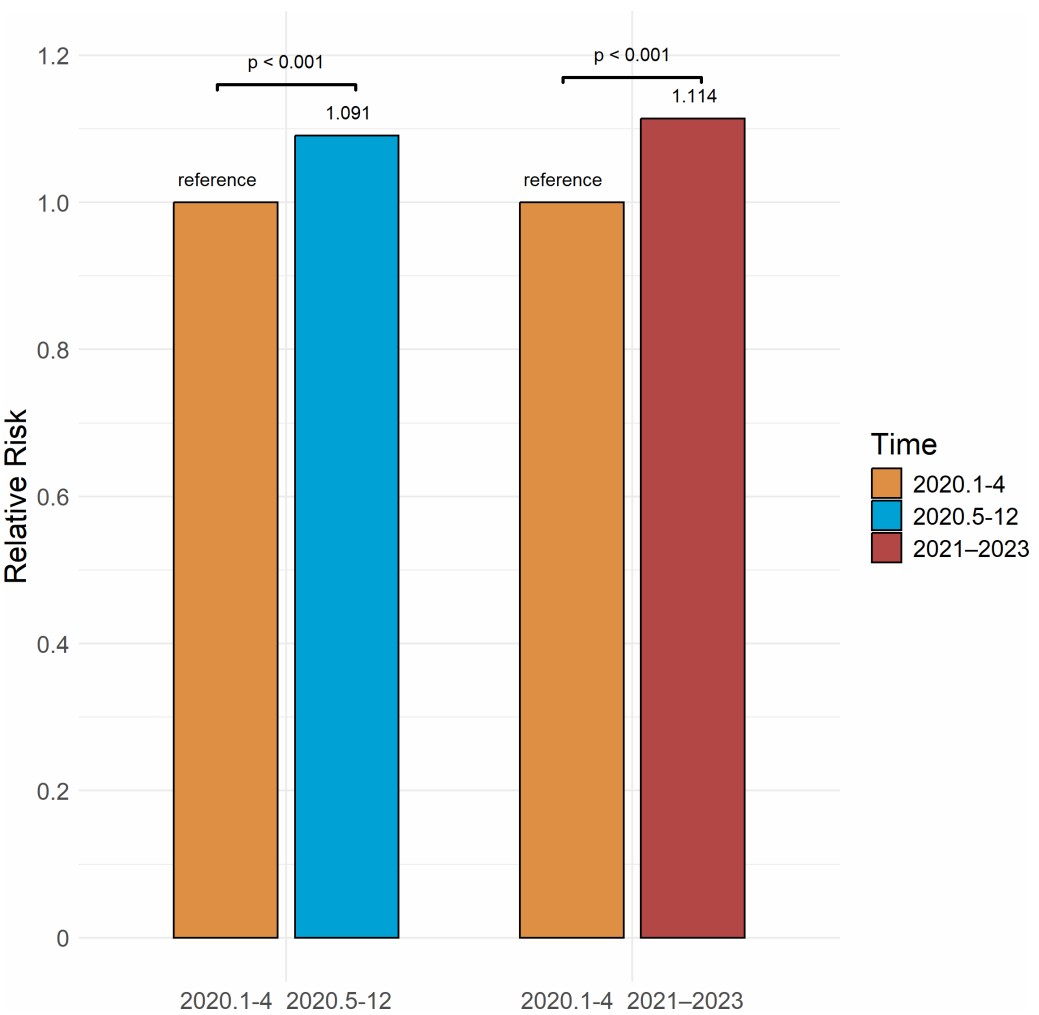

**Figure 4** Relative risk of depression among Chinese older adults 2020–2023.

a leave-one-out approach (Table S6). These findings further confirm the stability of our primary analysis.

## DISCUSSION

This meta-analysis included 101 studies published between 2017 and November 20, 2024, with data collected between 2017 and 2023. The results showed that the prevalence of depression among older adults in China was 25.8% (95% CI [21.7–29.9]) during 2017–2019 and 23.8% (95% CI [19.8–27.8]) during 2020–2023. To further investigate changes during the pandemic, we conducted an additional analysis focusing on 2020–2022, which showed a pooled prevalence of 23.1% (95% CI [19.2–27.0]). In 2023, the estimated prevalence was 33.9% (95% CI [20.3–47.6]) based on three studies ($n = 3$), but due to the limited number of studies available ($n = 3$), these findings should be interpreted with caution. These results

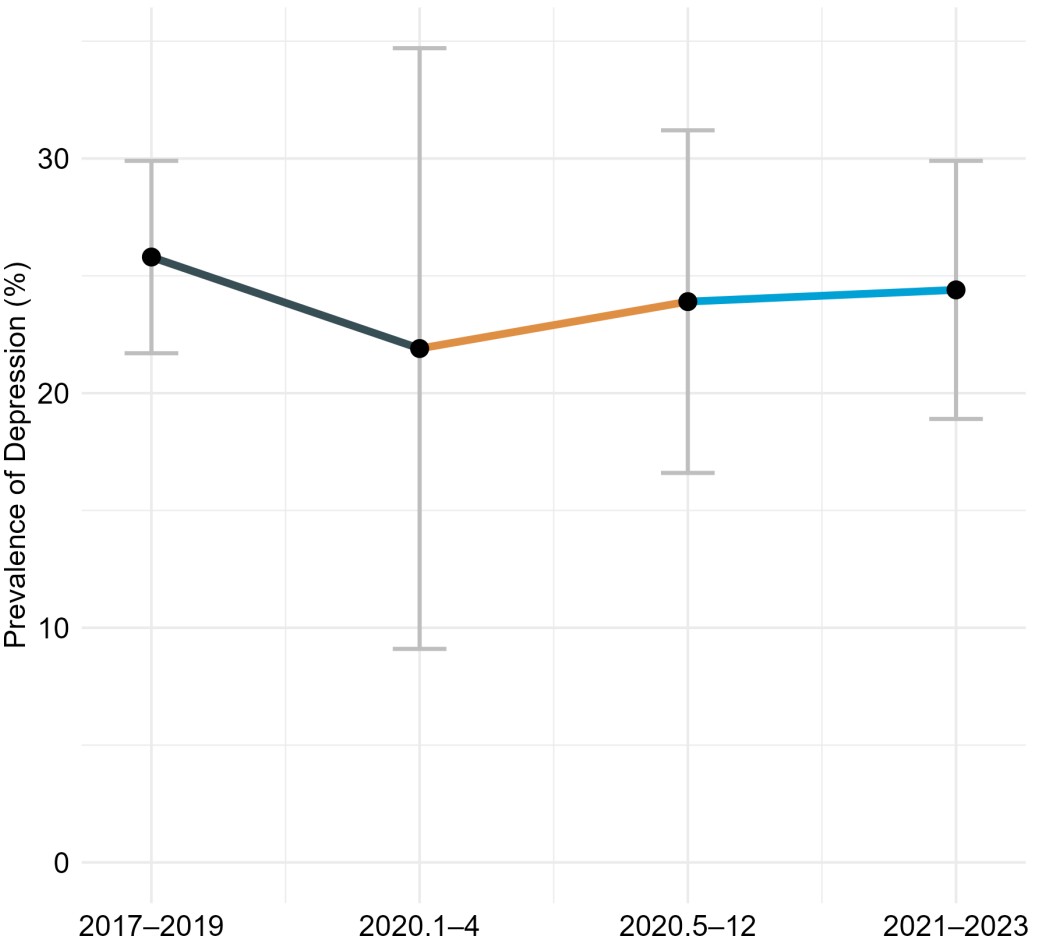

**Figure 5** Trend of the prevalence of depression among older adults in China before and during the COVID-19 pandemic.

suggest that the prevalence of depression fluctuated across different stages of the pandemic, requiring the need for further research.

Subgroup analyses showed significant variability in depression prevalence across sex, living area, education level, marital status, and residential status, all of which contributed to the observed high heterogeneity. Older women, rural residents, those with lower education levels, and individuals living alone showed consistently higher depression prevalence before and during the pandemic. The observed heterogeneity in this meta-analysis may come from the use of different measurement tools, variations in study samples, and China's unique pandemic response. Subgroup analysis helped explain some of these differences. However, future studies should use standardize depression assessment methods to make comparisons more accurate.

Compared to the pre-pandemic period (2017–2019), the prevalence of depression among urban older adults slightly increased during the pandemic period (2020–2023), while it slightly decreased among rural older adults. The increased infection risk and
**Table 1  Subgroup analysis.**

| Subgroup | 2017–2019 (pre-COVID-19) | | | | | 2020–2023 (during the pandemic) | | | | |
|---|---|---|---|---|---|---|---|---|---|---|
| | No. of studies | Pooled prevalence (95% CI) | Heterogeneity, I²(%) | Cochran's Q | p-value | No. of studies | Pooled prevalence (95% CI) | Heterogeneity, I²(%) | Cochran's Q | p-value |
| **Sex** | | | | | | | | | | |
| Male | 31 | 21.0 (17.8–24.3) | 99.2 | 639.8 | <0.001 | 22 | 20.7 (14.8–26.7) | 99.6 | 179.8 | <0.001 |
| Female | 32 | 26.6 (22.2–30.9) | 99.5 | | | 22 | 25.9 (18.2–33.7) | 99.7 | | |
| **Age group** | | | | | | | | | | |
| 60–69 | 19 | 24.3 (18.4–30.2) | 99.5 | | | 7 | 25.6 (6.7–44.4) | 99.8 | | |
| 70–79 | 16 | 27.4 (20.5–34.3) | 99.2 | 180.8 | <0.001 | 6 | 32.0 (7.3–56.8) | 99.7 | 5.6 | 0.059 |
| ≥80 | 13 | 31.0 (21.8–40.3) | 98.8 | | | 8 | 33.7 (18.1–49.4) | 97.9 | | |
| **Living area** | | | | | | | | | | |
| Rural | 11 | 26.5 (15.5–37.4) | 99.9 | 1500 | <0.001 | 8 | 26.4 (15.6–37.2) | 99.4 | 66.8 | <0.001 |
| Urban | 10 | 19.5 (14.6–24.4) | 99.4 | | | 6 | 20.0 (11.7–28.4) | 98.8 | | |
| **Education level** | | | | | | | | | | |
| Primary and below | 22 | 24.0 (19.3–28.7) | 99.5 | | | 14 | 31.0 (22.5–39.5) | 99.5 | | |
| Junior high school | 14 | 18.9 (14.2–23.6) | 98.6 | 794.8 | <0.001 | 9 | 23.5 (13.0–34.1) | 98.9 | 122.2 | <0.001 |
| Senior high school and above | 17 | 18.1 (14.6–21.5) | 97.7 | | | 9 | 17.9 (8.5–27.3) | 98.6 | | |
| **Marital status** | | | | | | | | | | |
| Single/Divorce/Widowed | 21 | 29.8 (23.7–35.8) | 99.3 | 181.7 | <0.001 | 16 | 32.5 (23.0–42.0) | 99.0 | 489.7 | <0.001 |
| Married | 21 | 23.6 (19.0–28.2) | 99.6 | | | 16 | 22.0 (14.7–29.3) | 99.6 | | |
| **Residential status** | | | | | | | | | | |
| Live alone | 17 | 29.7 (23.0–36.4) | 98.5 | 319.8 | <0.001 | 8 | 36.0 (20.0–52.1) | 96.9 | 117.2 | <0.001 |
| Not live alone | 17 | 23.0 (18.9–27.0) | 99.6 | | | 8 | 17.7 (10.7–24.6) | 98.8 | | |

vulnerability in urban areas likely lead to increased levels of psychological stress (*Liu et al., 2021b*). Previous study has suggested that people living in urban communities are more likely to contract COVID-19 and face a higher risk of poor psychological adaptation (*Wang et al., 2023a*). However, our study found that during the pandemic, the prevalence of depression among older adults was higher in rural areas than in urban areas. This result is consistent with previous research on rural–urban differences in depression (*Wu et al., 2023*). Compared to older adults in urban, rural older people were more affected by increased social isolation (*Kaur & Rani, 2024*), difficulties in accessing medical resources (*Gautam et al., 2020*; *Yang et al., 2020*), economic pressures caused by the pandemic, gaps in information access (*Cheshmehzangi, Zou & Su, 2022*), and disparities in mental health support systems between rural and urban areas. Older adults with higher education levels (senior high school and above) showed lower prevalence of depression, likely due to a better understanding of COVID-19 prevention and health information (*Abdelhafiz et al., 2020*; *Yue et al., 2021*), consistent with previous studies (*Li et al., 2014*). Conversely, older adults living alone exhibited a significantly higher prevalence of depression, due to increased loneliness, limited access to information, and challenges in accessing healthcare resources,

as corroborated by earlier studies (*Bu, Steptoe & Fancourt, 2020*). In contrast, those not living alone likely experienced reduced loneliness and anxiety due to family support and companionship (*Torres, Oliver & Tomás, 2023*).

Our findings are similar to previous studies. One study reported a depression prevalence of 25.55% from 2010 to 2019 (*Rong et al., 2020*), while another found 23.94% in 2020 (*Wu et al., 2023*). Our study shows that depression prevalence changed during different stages of COVID-19. It first declined in early 2020, then gradually increased from mid-2020 to 2023. This pattern is similar to global findings but also has some unique characteristics.

Before the pandemic, depression prevalence among older adults in China was close to international estimates. A meta-analysis on depression in infectious diseases, including COVID-19, reported a pooled prevalence of 26.0% (*Yuan et al., 2022*). This is similar to the pre-pandemic prevalence in this study. Furthermore, global studies found that older women, rural residents, single or widowed individuals, and those living alone were at a higher risk of depression (*Yuan et al., 2022*). A UK study on older adults in London during the early COVID-19 period reported that single, divorced, or widowed individuals experienced an increase in depression after lockdown (*Robb et al., 2020*), aligning with findings from Spain (*Rodríguez-Rey, Garrido-Hernansaiz & Collado, 2020*) and the US (*Ettman et al., 2020*).

During the early stage of the pandemic (January–April 2020), the prevalence of depression among elderly adults significantly declined compared with 2017–2019 levels. This finding is different from some studies in other countries, where depression prevalence rose right after the COVID-19 outbreak. For example, a study in England found that depression prevalence increased from 12.5% before the pandemic to 22.6% in mid-2020, then went up to 28.5% by late 2020 (*Zaninotto et al., 2022*). In Ireland, another study reported that depression prevalence among older adults reached 19.8% in the early months of the pandemic (*Briggs et al., 2021*). However, findings from The Netherlands found that older adults' mental health stayed mostly the same at the start of the pandemic (*Van Tilburg et al., 2021*). This suggests that differences between countries may have played a role. The period (January–April 2020) marked China's emergency COVID-19 response phase (*Chen et al., 2021*), with strict measures including lockdowns, movement restrictions, home isolation, and mandatory mask use (*Cheng et al., 2023*). These measures increased public health awareness, effectively controlled virus transmission (*Ayouni et al., 2021*; *Murphy et al., 2023*), and reduced infection risk for older adults (*Lai et al., 2020*; *Qi et al., 2022*). Additionally, the timely release of information by the Chinese government helped alleviate public health concerns and stabilize emotions (*Wang et al., 2020a*; *Dai et al., 2020*). During lockdowns and work suspensions, families staying at home provided increased of support, offering psychological comfort and care to older adults and helping them cope with pandemic-related stress and uncertainty (*Arpino et al., 2022*). Early strict control measures were shown to be effective for epidemic prevention (*Anderson et al., 2021*; *Xiao et al., 2021*), not only in controlling the spread of the virus but also in alleviating psychological stress and depression. However, this finding may be affected by the small number of studies included for this period. The limited data could have influenced the pooled prevalence

estimate. More research is needed to see if this decline was a real drop in depression or a result of the available studies.

As the pandemic progressed, between May and December 2020, the prevalence of depression among older adults in China slightly increased but remained below pre-pandemic levels. In England, a study using a nationally representative sample found that depression prevalence continued to rise throughout 2020. During this period, as the pandemic was initially controlled, policies shifted to regular epidemic prevention and control (*Liang et al., 2021b*). People were still encouraged to maintain social distancing, good health practices, and reduce outdoor activities, thus extending the health benefits of early interventions (*Qi et al., 2022*). However, the prolonged and recurrent nature of the pandemic, along with its uncertainties, posed ongoing high infection and health risks for the older adults (*Zhang et al., 2020*). Although the spread of smartphones and the internet has improved information access, the prevalence of fake news and inaccurate information on social media and short video platforms has exacerbated health concerns among older adults (*Yue et al., 2021*; *Irwin, 2020*). This information uncertainty, coupled with chronic diseases and other health issues common in older adults likely exacerbates their depressive symptoms (*Dubey et al., 2020*). Although social restrictions were somewhat eased, the continued spread of the virus limited older adults' participation in social activities (*Armitage & Nellums, 2020*; *Torres, Oliver & Fernández, 2024*), potentially leading to feelings of loneliness and helplessness. Thus, compared to early pandemic stage, depression prevalence increased slightly during this period.

Between 2021 and 2023, the prevalence of depression among older adults continued to rise, although it remained below the pre-pandemic levels. New virus variants and fluctuating waves of the pandemic increased public uncertainty and fear (*Arpino et al., 2022*). In August 2021, China's dynamic zero-COVID-19 policy, which included frequent local lockdowns and nucleic acid testing, severely disrupted daily life (*Wang & Huang, 2022*). The switch between strict isolation and eased social distancing further worsened depression among older adults (*Phiri et al., 2021*). Additionally, the dynamic zero-COVID-19 policy increased economic and personal burdens, leading to a decline in quality of life and greater psychological distress among older adults (*Bai et al., 2022*; *Losada-Baltar et al., 2021*). However, as restrictions eased, social interactions and healthcare services resumed, which contributed to a gradual stabilization in the prevalence. These findings show that early pandemic measures may have temporarily lowed depression risk. As the pandemic went on, stress increased, and depression prevalence rose among older adults.

Our study shows that older adults need better social support and welfare programs to help prevent depression (*Tang, Jiang & Tang, 2021*). This is especially important for those in rural areas and those who live alone. The Chinese government should make mental health services easier to access in communities. One way is to add mental health checkups to regular doctor visits. Another way is to train healthcare workers to identify and support people with mental health problems. Communities should build support networks, teach people about mental health, and reduce the stigma around depression. Volunteers and social workers can visit older adults who live alone or are at high risk to provide help and companionship. In rural areas, families and local communities give strong support. These

networks should be used to improve mental health care. In cities, older adults who live alone may not have the same support. Online doctor visits and other technology can help fill this gap. During the pandemic, depression levels changed at different times. This shows that mental health support should be part of future emergency plans. The government, communities, hospitals, and non-government organizations should work together to create long-term mental health programs. These efforts will help older adults get the care they need and improve their well-being.

This study has some limitations. First, the small number of studies from 2023 made it hard to examine long-term mental health trends after pandemic. Second, high heterogeneity was observed among studies. Different screening tools, cutoff scores, and assessment methods may have affected prevalence estimates. There was also a wide range of depression assessment tools. While some tools, such as GDS, PHQ, and CES-D, were commonly used, others appeared in only few studies. Because of this, we could not perform a detailed subgroup analysis based on assessment methods. Third, due to the limited number of studies available for each demographic subgroup (*e.g.*, sex, living area, education level, marital status, and residential status) across different pandemic stages, we could not calculate pooled prevalence estimates for specific demographic trends over time. Future research should use standardized depression measurements tools and include larger, more representative datasets to facilitate more precise trend analysis.

## CONCLUSION

The study revealed that the prevalence of depression among older adults in China was 25.8% (95% CI [21.7–29.9]) from 2017 to 2019 and 23.8% (95% CI [19.8–27.8]) from 2020 to 2023. The effect of the COVID-19 pandemic on this prevalence has changed across stages. Compared with that from 2017–2019, the prevalence significantly decreased between January and April 2020. From May to December 2020, it slightly increased but remained below pre-pandemic levels. By 2021–2023, the prevalence had risen and remained below the pre-pandemic levels. We observed a gradual increase in the prevalence of depression among older adults in China during the COVID-19 pandemic. These changes highlight the complex impact of the pandemic and its control measures on mental health. Early strict control measures were associated with reduced depression levels and helped shield older adults from the immediate health risks of the virus, extended periods of social isolation, uncertainty surrounding the pandemic, and frequent policy adjustments, which may have contributed to increased mental health challenges in the long term. Future research should continue to monitor post-pandemic mental health trends and explore strategies to strengthen psychological resilience among older adults.

### Funding

This study was supported by The National Social Science Fund of China (20BRK020) and Economic & Social Research Council (ESRC) (No. ES/P011055/1). The funders had no

role in study design, data collection and analysis, decision to publish, or preparation of the manuscript.

## Grant Disclosures
The following grant information was disclosed by the authors:
The National Social Science Fund of China: 20BRK020.
Economic & Social Research Council (ESRC): No. ES/P011055/1.

## Competing Interests
The authors declare there are no competing interests.

## Author Contributions

- Xin Zhao conceived and designed the experiments, performed the experiments, analyzed the data, prepared figures and/or tables, authored or reviewed drafts of the article, and approved the final draft.
- Xiaojing Du conceived and designed the experiments, authored or reviewed drafts of the article, and approved the final draft.
- Shuliang Bai performed the experiments, analyzed the data, prepared figures and/or tables, authored or reviewed drafts of the article, and approved the final draft.
- Pianpian Zheng conceived and designed the experiments, authored or reviewed drafts of the article, and approved the final draft.
- Xun Zhou performed the experiments, authored or reviewed drafts of the article, and approved the final draft.
- Zhenjie Wang conceived and designed the experiments, performed the experiments, analyzed the data, prepared figures and/or tables, authored or reviewed drafts of the article, and approved the final draft.

## Data Availability
This is a systematic review/meta-analysis.

## Supplemental Information
Supplemental information for this article can be found online at http://dx.doi.org/10.7717/peerj.19251#supplemental-information.

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
