# Peer review of "Differences in depression prevalence among older adults in China before and during the COVID-19 pandemic: a systematic review and meta-analysis"

_PeerJ, doi:10.7717/peerj.19251_

## Round 0.1 · original submission · Major Revisions

I am recommending major revision because I hope you are going to satisfactorily address all comments from reviewers and also my general comments. It is important that you have registered in Prospero. Also reply to the following comments:

brief statement on the statistical significance of findings (e.g., RR and p-values) and explicitly highlight the public health implications for intervention strategies targeting elderly mental health in China.

Refine the problem statement to emphasize the lack of systematic evaluations of depression prevalence trends during the COVID-19 pandemic and clearly articulate the study's aim to fill this gap.

Specify the rationale for the time range (2017–2024) and ensure details about inclusion/exclusion criteria (e.g., age, validated tools) are consistently presented. Confirm the PRISMA diagram includes accurate data flow and is visually clear.

Results must provide a more detailed explanation of heterogeneity sources in subgroup analyses and integrate additional figures or tables summarizing trends by demographics (e.g., sex, rural/urban residence).

In the discussion, address potential biases due to diagnostic tool variability and align findings with global depression trends during COVID-19. Emphasize the need for targeted mental health interventions for vulnerable subgroups like rural residents and those living alone.

Reviewer 1 ·

Basic reporting

Thank you for preparing this article. This manuscript has some comments:
1. Replace the word elderly with older adults
2. Poor quality of prisma figure
3. For the inclusion & exclusion criteria, `in cases of multiple, publications based on the same dataset, only the study with the most comprehensive data was included`. What does the meaning of this sentence?
4. Reference for quality assessment
5. Please provide the Prospero registration trial number.

Experimental design

The final stage of the prisma flowchart is confusing. Why were there three stages ?
-What were the study design of the finally selected article?
-The duration of the selected study ranged from pre-covid to post-Covid, however the discussion were not structured based on these phases.

Validity of the findings

-the discussion of the article must focus on the pre, during and post covid phases based on the article selected.
-the trial must be registered with Prospero

·

Basic reporting

It has been a pleasure reviewing your article. We have found it very interesting and as important new information to the area of research. Nevertheless, we have some minor comments to your study:

L.51 – 52: ‘Depressive disorder is a highly prevalent psychiatric condition that is characterized by prolonged periods of depressed mood or diminished interest in previously enjoyed activities’ is deficient regarding the description of what depressive disorder means to the individuals everyday and how depressive disorder affects the psychological well-being of an individual. Please find more evidence to support your description/ definition of this area.

In your RESULT paragraph we mean that the main results should be presented at the beginning of the paragraph. So that ‘Published evidence characteristics’ and ‘Subgroup analysis’ will be presented later. We thereby need to see a better structured RESULT paragraph.

There is a problem with the sentence l. 165 – 167 regarding when the studies were conducted and when the studies were collected. A better explanation is needed here in terms of time periods.

In l. 282 the word ‘relaxed’ is used to describe a less strict social restrictions regarding COVID 19. Please use a more suitable scientific word instead.

Experimental design

No further comments.

Validity of the findings

In your analysis we are concerned about the large time interval. We would like to see a sensitivity analysis where the period with ‘active’ COVID 19 is smaller. In our opinion this time interval should be from January 2020 to December 2022. The shorter time period could be conducted as a sensitivity analysis.

Additional comments

Nora Kathrine Gylling has been part of the review process too.

---

## Round 0.2 · accepted · Accept

Thanks for addressing all the comments thoroughly, and I am happy to recommend your article for publication. The reference style and grammar need to be checked before publication.

·

Basic reporting

Fine

Experimental design

Appropriate

Validity of the findings

Great

Additional comments

The authors have addressed all our comments and concerns from the first review.